# Diagnostic Accuracy of High-Grade Intraepithelial Papillary Capillary Loops by Narrow Band Imaging for Early Detection of Oral Malignancy: A Cross-Sectional Clinicopathological Imaging Study

**DOI:** 10.3390/cancers14102415

**Published:** 2022-05-13

**Authors:** Airi Ota, Ikuya Miyamoto, Yu Ohashi, Toshimi Chiba, Yasunori Takeda, Hiroyuki Yamada

**Affiliations:** 1Department of Oral and Maxillofacial Reconstructive Surgery, Division of Oral and Maxillofacial Surgery, School of Dentistry, Iwate Medical University, Morioka 020-8505, Japan; ootaa@iwate-med.ac.jp (A.O.); yohashi@iwate-med.ac.jp (Y.O.); yamadah@iwate-med.ac.jp (H.Y.); 2Department of Oral Medicine, Division of Internal Medicine, School of Dentistry, Iwate Medical University, Morioka 020-8505, Japan; toschiba@iwate-med.ac.jp; 3Department of Oral and Maxillofacial Reconstructive Surgery, Division of Clinical Pathology, School of Dentistry, Iwate Medical University, Morioka 028-3695, Japan; ytakeda@iwate-med.ac.jp

**Keywords:** oral potentially malignant disorders, oral squamous cell carcinoma, intraepithelial papillary capillary loops, early diagnosis, narrow band imaging

## Abstract

**Simple Summary:**

This study aimed to clarify the advantages and disadvantages of conventional visual inspection (CVI), endoscopic white light imaging (WLI), and narrow-band imaging (NBI) and to examine the diagnostic accuracy of intraepithelial papillary capillary loops (IPCL) for the detection of oral squamous cell carcinoma (OSCC). All OSCC cases showed high-grade (Type III–IV) IPCL. A non-homogeneous lesion with high-grade IPCL strongly suggested malignancy. Our results indicate that WLI and NBI are powerful tools for detecting precancerous and cancerous lesions using IPCL. However, NBI is influenced by mucosal thickness; therefore, image interpretation is important.

**Abstract:**

This study aimed to clarify the advantages and disadvantages of conventional visual inspection (CVI), endoscopic white light imaging (WLI), and narrow-band imaging (NBI) and to examine the diagnostic accuracy of intraepithelial papillary capillary loops (IPCL) for the detection of oral squamous cell carcinoma (OSCC). This cross-sectional study included 60 participants with oral mucosal diseases suspected of having oral potentially malignant disorders (OPMDs) or OSCC. The patients underwent CVI, WLI, NBI, and incisional biopsy. Images were evaluated to assess the lesion size, color, texture, and IPCL. Oral lichen planus (OLP) and oral leukoplakia lesions were observed in larger areas with NBI than with WLI; 75.0% were associated with low-grade (Type 0–II) IPCL. Various types of oral leukoplakia were seen; however, all OSCC cases showed high-grade (Type III–IV) IPCL. The diagnostic accuracy of high-grade IPCL for OSCC showed a sensitivity, specificity, positive predictive value, negative predictive value, and accuracy of 100%, 80.9%, 59.1%, 100%, and 85.0%, respectively. A non-homogeneous lesion with high-grade IPCL strongly suggested malignancy. Overall, our results indicate that WLI and NBI are powerful tools for detecting precancerous and cancerous lesions using IPCL. However, NBI is influenced by mucosal thickness; therefore, image interpretation is important for accurate diagnosis.

## 1. Introduction

Oral squamous cell carcinoma (OSCC) is an aggressive tumour with poor prognosis [1,2,3]. Early diagnosis and treatment of malignancies usually improve long-term cure and survival [4,5]. Clinically, oral cancer is preceded by a premalignant state and accurately diagnosed by incisional biopsy and histopathological examination [5,6].

This premalignant concept has recently been expanded, and these disorders are listed under the umbrella term oral potentially malignant disorders (OPMDs) [4,6,7,8]. There are several diseases categorized as OPMDs; oral lichen planus (OLP) and oral leukoplakia are the most frequently encountered lesions. In particular, OLP is common and affects approximately 1–2% of the general population [9,10,11]. Comprehensive inspection using tools is thought to improve the diagnostic accuracy [12]. With the development of instruments, endocytoscopy has enabled the pathological diagnosis of oral malignancies in situ and has allowed the observation of both structural and cytological atypia [13].

Preoperative examinations using white light imaging (WLI) endoscopy may not agree with postoperative biopsy results, which leads to overtreatment or undertreatment in some cases [1,14,15]. Consequently, the main recommendations for OPMDs and OSCC are incisional biopsy and histopathological diagnosis [14]. Conversely, narrow band imaging (NBI) allows for the visualization of capillary patterns of the submucosal layer among epithelial papillae (intraepithelial papillary capillary loops; IPCLs) [16]. According to systematic reviews, NBI, alongside WLI, is an important technique that improves the diagnostic accuracy of head and neck cancers [17,18]. NBI has been shown to be effective in the detection of dysplastic/malignant lesions of the upper aerodigestive tract [19,20,21]. These images are influenced by several factors; however, there are few studies on the limitations of CVI, WLI, and NBI in the field of oral malignancies. In addition, it is also unclear how valuable IPCL is in the diagnosis of oral cancerous lesions. The purpose of this study was to clarify the advantages and disadvantages of CVI, WLI, and NBI and to examine the diagnostic accuracy of IPCL for the detection of OPMDs and cancerous lesions along with their pathological outcomes.

## 2. Materials and Methods

This single-center, non-interventional, cross-sectional study was conducted at the Division of Oral and Maxillofacial Surgery, Iwate Medical University, from 1 February 2021 to 31 February 2022 in accordance with the principles stated in the Declaration of Helsinki (1964) and its later amendments. Iwate Medical University is an urban university hospital that serves approximately 1,200,000 people in the Iwate prefecture, and the Division of Oral and Maxillofacial Surgery treats approximately 3000 new patients and 15,000–16,000 revisiting patients annually. The study design was approved by the Ethics Committee of the Faculty of Dentistry at Iwate Medical University (01327).

This study was conducted in accordance with the STROBE and STARD guidelines [22,23]. Written informed consent was obtained from all the participants.

The inclusion criteria were: the presence of OPMDs or cancerous lesions, and no previous procedure (surgery, chemotherapy, and/or radiotherapy). For the observation of normal mucosa, healthy volunteers who consented to the study were included. All consecutive patients presenting to the Division of Oral and Maxillofacial Surgery with the chief complaint of mucosal lesions suspected to be OPMDs or OSCC were included. The exclusion criteria were: patients on strong anticoagulant therapy; patients with severe heart, lung, or renal dysfunction; or weak patients who could not undergo resection by incisional biopsy or surgery.

### 2.1. Participants

All patients who visited the Division of Oral and Maxillofacial Surgery at Iwate Medical University and were categorized as cases of suspected OPMDs or OSCC (*n* = 95) were considered potentially eligible.

After several clinical examinations, 15 participants were excluded because they had different diseases. The remaining 80 participants were clinically diagnosed with OPMDs or OSCC. Twenty participants were excluded because they did not provide consent for incisional biopsy. Thus, 60 participants were finally included in the study. Figure 1 shows the flowchart for study inclusion.

### 2.2. Treatment of Patients

Initially, participants were examined by CVI, and the total area of lesions was scored using the classification of anatomical sites and subsites of the oral cavity (Figure 2) [24]. When each area was regarded as one point and the lesion crossed multiple areas, the total number of points was defined as the size of the lesion. The lesions were examined using WLI and NBI, and the total lesion area was recorded. All oral regions including the hard palate, upper gingiva, buccal mucosa, lower gingiva, dorsum of the tongue, lateral tongue, oral floor, and oropharynx were observed. WLI endoscopic examinations were performed initially with standard illumination in a wide view to observe the whole lesion and its surrounding mucosa, and then magnification was increasingly applied until the capillaries could be analyzed in detail. The same procedure was performed for NBI illumination. For the control volunteers, the same sites of the normal mucosa were screened. After observation by WLI and NBI, patients were considered fit for a full-thickness or partial-thickness biopsy in areas where the lesions were suspected to be the worst. Biopsies were not performed in the volunteers.

The examinations were performed using an Evis Lucca Spectrum Video Imaging System (CV-190 processor and CLV-190 light source; Olympus Medical Systems Corp., Tokyo, Japan). The system contains standard and NBI filters that can be changed during examinations by using a button on the keyboard. The rotating interference red–green–blue (RGB) narrowband filter is interposed after the xenon light source, where the light passes through and narrows the bandwidth, changing the spectral characteristics of the incident light [25]. The optical fibers have center wavelengths of 500 nm, 445 nm, and 415 nm corresponding to penetration depths of 240 µm, 200 µm, and 170 µm, respectively [26,27]. The light penetration depth depends on the wavelength, likely because of the absorption and scattering processes that occur in the tissue structures [28]. Short wavelengths scatter easily and are selectively absorbed by hemoglobin, thus providing good contrast for the mucosal microvasculature, especially the capillary loops. Longer wavelengths scatter minimally and penetrate more deeply. Each reflected light spectral feature was captured by a charge-coupled device chip at the tip of the endoscope, and a unique high-resolution contrast image was reconstructed [29].

Overall, the participants included 10 normal volunteers, 16 OLP cases, 31 clinical oral leukoplakia cases, and 13 OSCC cases, and the oral mucosa was comprehensively observed using CVI, WLI, and NBI. The patients included 27 men and 33 women (mean age, 62.7; range, 34–87 years) (Table 1). Information about the individual participants was noted as Appendix A.

### 2.3. Clinical Subtype of the Lesions

The clinical subtypes of the lesions in OLP, clinical oral leukoplakia, and OSCC were compared and examined. The WLI images were recorded and transferred to a computer digitally. According to Jäwert et al., the clinical subtypes of the classified lesions can be divided into two subcategories: homogeneous and non-homogeneous [2]. Homogenous lesions are defined as well-demarcated lesions with a uniformly white, plaque-like appearance and a flat surface. A more speckled appearance with irregular red and white areas and/or a nodular or verrucous surface is indicative of a non-homogeneous lesion. Typical homogeneous and non-homogeneous lesions are shown in Figure 3a,b.

### 2.4. IPCL Classification

Based on the classification by Vu et al., the IPCL classification of the oral mucosa was conducted by modifying Inoue’s classification, which consists of four progressive increases in the IPCL pattern type with a 5th (type 0) when IPCL patterns are not visualized because of thick keratosis or oral leukoplakia: type I (physiological arborization of IPCL), type II (meandering or dilated IPCL), type III (convoluted/winding and/or elongated IPCL), and type IV (complete loss of organization/annihilation of the IPCL) (Figure 4a–e) [30,31].

### 2.5. Statistical Analysis

Considering the exploratory nature of the research and absence of significant data on the use of instruments in this setting, no preliminary power analysis was performed. Continuous variables were recorded as the mean ± standard deviation. Statistical comparisons were performed using Fisher’s exact test, Wilcoxon’s rank sum test with Bonferroni correction, and the Kruskal–Wallis test, which set the level of significance at 0.05 per number of comparisons.

To determine the reliability of the IPCL classification and to avoid observer bias, all IPCL images in this study were recorded separately in random order by two trained independent observers (A.O. and I.M.). Intra- and inter-observer (A.O. and I.M.) reliability was assessed among the separate obtained measurements to eliminate memory bias. Cohen’s kappa value for IPCL classification was determined as the degree of congruence; a value >0.80 was indicative of good congruence. All images were examined twice in a 21-day interval. For the diagnostic accuracy of type III or IV of IPCL for predicting OSCC, sensitivity (Se), specificity (Sp), positive predictive value (PPV), negative predictive value (NPV), and accuracy (Ac) were calculated with 95% confident intervals (CI). Statistical analyses were performed using EZR (Saitama Medical Centre, Jichi Medical University, Saitama, Japan), a graphical user interface for R (The R Foundation for Statistical Computing, Vienna, Austria) [32].

## 3. Results

### 3.1. Normal Tissue Characteristics

Examination of normal volunteers revealed that it was difficult to observe superficial capillaries on the dorsum of the tongue, hard palate, and gingiva owing to the thickness of the mucous membrane and all were classified as type 0 IPCL. In contrast, the buccal mucosa and lips, where blood vessels were visible, were all type I IPCL. Table 2 shows the site-specific IPCL classification of the normal intraoral mucosa in the volunteers (Table 2).

### 3.2. Comparison of the Extent of Lesions between the CVI, WLI, and NBI

When comparing the extent of the lesion according to the visualization method, the OLP was found to be 2.9 ± 1.9 for CVI, 4.0 ± 2.3 for WLI, and 5.1 ± 2.8 points for NBI (Figure 5a); there were significant differences between the imaging modalities. Clinical oral leukoplakia was 1.9 ± 1.7 for CVI, 3.0 ± 2.3 for WLI, and 3.5 ± 2.6 points for NBI (Figure 5b); there were significant differences between the groups. OSCC was 1.4 ± 0.6 for CVI, 1.4 ± 0.6 for WLI, and 1.4 ± 0.6 points for NBI (Figure 5c); there were no significant differences between the groups. For OLP and clinical oral leukoplakia, a wide range of lesions could be identified using WLI or NBI, whereas for OSCC, the range did not change with the use of instruments.

### 3.3. Locations and Textures of the Lesions

The locations and appearances of the lesions were observed. The OLP lesions of 14 (87.5%) patients were located at the buccal membrane. There were statistically significant differences between groups (Fisher’s exact test, *p* < 0.0001). The OLP demonstrated a white patch in the homogeneous red inflammatory mucous membrane. Twenty-six (83.9%) of the clinical oral leukoplakia were homogeneous, while five (16.1%) were not. There were statistically significant differences between groups (Fisher’s exact test, *p* < 0.0001). The margins of the lesions were relatively well defined. Thirteen (100%) OSCC lesions were non-homogeneous. There were statistically significant differences between groups (Fisher’s exact test, *p* < 0.0001). Nine (69.2%) OSCC lesions were found on the lower lateral side of the tongue. Furthermore, there were red lesions within the thick keratinized surfaces of the OSCC lesions. In other words, red lesions were observed in areas where the white lesions had been torn. The margins of the lesions were relatively well defined.

### 3.4. IPCL

The IPCL showed favorable reliability in terms of Cohen’s Kappa index. Intra-observer reliability was 0.81 (95% CI: 0.69–0.93), and inter-observer reliability was 0.75 (95% CI: 0.61–0.88) (Table 3). Twelve (75.0%) OLP lesions were low-grade (I–II) IPCL patterns, and four lesions (25%) were type III IPCL (Figure 6a). Clinical oral leukoplakia showed various type of IPCL including nine (29.0%) type 0; 14 (45.1%) type I; three (9.7%) type II; three (9.7%) type III; and two (6.7%) type IV (Figure 6b). Owing to the thickness of the oral leukoplakia, it was difficult to observe the IPCL directly. IPCLs were therefore observed at the borders of the lesions. There were five (16.1%) non-homogeneous lesions for all clinical oral leukoplakia types, 80% of which had type III or IV IPCL. For pathological diagnosis, the three type III and two type IV IPCL showed moderate epithelial dysplasia and proliferative verrucous leukoplakia, respectively. OSCC was composed of five (38.5%) type III IPCL and eight (61.5%) type IV IPCL. OSCC showed a clearly advanced IPCL pattern (Figure 6c). The distribution of IPCL varied according to disease. There were statistically significant differences (Kruskal–Wallis test, *p* < 0.001). Depending on the site of the mouth, the loops were found in parallel or perpendicular locations in relation to the surface. The IPCL appeared as scattered brown dots in the perpendicular position, while they appeared as waved lines in parallel positions (Figure 4e). Thus, types III and IV IPCL are related to the malignancy of the lesion

Considering the pathological data, the diagnostic accuracy of type III or IV IPCL in the prediction of OSCC was investigated. The results of types III and IV IPCL by NBI inspection for the entire histopathological outcomes showed that the sensitivity (Se), specificity (Sp), positive predictive value (PPV), negative predictive value (NPV), and accuracy (Ac) were 100%, 80.9%, 59.1%, 100%, and 85.0%, respectively. The results are summarized in Table 4.

## 4. Discussion

Endoscopy-assisted WLI inspection is useful, and NBI is an influential tool for the early detection of oral cancerous lesions, as it focuses on the grade of IPCL. In contrast, in the thick keratinized mucosa, blood vessels cannot be seen using NBI. This is a clear limitation of the instrument.

NBI observing the surface layer depicts the lesions very well. OLP is frequently observed in the buccal mucosa or lower lip due to the characteristics of the light; OLP is only recognized as a lesion when visualization is possible [33]. OLP is an immunological inflammatory reaction that affects areas larger than expected. Considering the principle of NBI, lesions may spread diffusely. The wide range of inflammation in OLP implies that one of the most important characteristics of OSCC arising in OLP lesions is their tendency to be multifocal, according to the concept of field cancerization [34,35]. This implies that with only visual inspection, OLP lesions were observed in only a small portion of the entire lesion, and the site of tumorigenesis may have been underdiagnosed; therefore, NBI would be more useful to diagnose the spread of OLP lesions compared with CVI or WLI.

Regarding the surface texture of the lesions, OLP and OSCC lesions tended to be non-homogeneous. However, there were differences between the groups. The non-homogeneous lesions of OLP had a background of an inflammatory red reaction with keratinized white mucosal lesions. Non-homogeneous OSCC lesions had a background of neoplastic white lesions with red mucosal lesions. Red lesions were observed in areas where white lesions had been torn. Redness reflects an inflammatory reaction on OLP, while indicating tumorigenesis and uncreative lesions in OSCC; therefore, non-homogeneous lesions of clinical oral leukoplakia are suggestive of the development of OSCC [2].

Regarding the IPCL analysis, type III IPCL was found in 25% of OLP and appeared to indicate inflammatory vascular change. Type III or IV tumors have an increased likelihood of displaying high-grade dysplasia and OSCC. The five patients with high-grade IPCL in clinical oral leukoplakia included three with moderate epithelial dysplasia and two with proliferative verrucous leukoplakia. These histological results indicate a high risk of OSCC. These results suggest that tumour angiogenesis might precede epithelial malignant transformation, as proliferative verrucous leukoplakia tends to undergo malignant transformation [36].

As seen in normal tissue, thick keratinized tissue is often difficult to find by IPCL; thus, a specialized technique is required. Homogeneous lesions generally show IPCL at the margin between the normal and abnormal mucosa. In contrast, non-homogeneous lesions frequently show a thin or ulcered mucosa, and the probability of detection of IPCL in this area is relatively high. Additionally, the relationship between toluidine blue, iodine staining, and IPCL was not examined in this study but would be worth investigating in the future [37].

The presence of type III or IV lesions indicates the possibility of malignancy. NBI can guide clinicians in selecting the best biopsy site based on the area with the most severe IPCL pattern, thereby minimizing the need for multiple biopsies [38]. NBI cannot replace histological evaluation, but careful observation may confirm a high degree of IPCL. In the future, it is expected that difficult to diagnose lesions will be a way to measure the rate of mutation in the genome [38].

This study has several limitations. First, only a small part of the lesions was excised for diagnosis, which may have resulted in sampling mistakes. Second, this research was a cross-sectional study and included a relatively small cohort of participants. A larger prospective multicenter study should be performed to confirm these data. Another limitation that needs to be stressed is the lack of a commonly accepted definition of “expert” and “experienced” in NBI procedures. This could be a potential source of bias. Clinical subtypes of the lesions and IPCL judgments are subjective and may contain measurement bias and observer bias. In our case, the operator (AO and IM) had performed examinations on at least 300 sites per year. However, further studies on operators’ learning curves are required to avoid the risk of major bias when assessing NBI reliability. Moreover, as diagnostic percentage standards are widely shown in the literature, self-evaluation could be achievable by comparing the operator’s NBI pattern evaluation with histopathological analysis [28].

## 5. Conclusions

Endoscopy-assisted visual inspection is useful for the detection of oral mucosal diseases, and NBI is a powerful tool for the early detection of oral cancerous lesions. Although OLP was detected in a relatively small area with only CVI, WLI and NBI can detect lesions in larger areas. Moreover, high-grade IPCL is a useful marker for the detection of OSCC; however, the NBI image is influenced by the thickness of the lesion, and interpretation of the image is important for understanding the principle of the instrument.

## Figures and Tables

**Figure 1 cancers-14-02415-f001:**
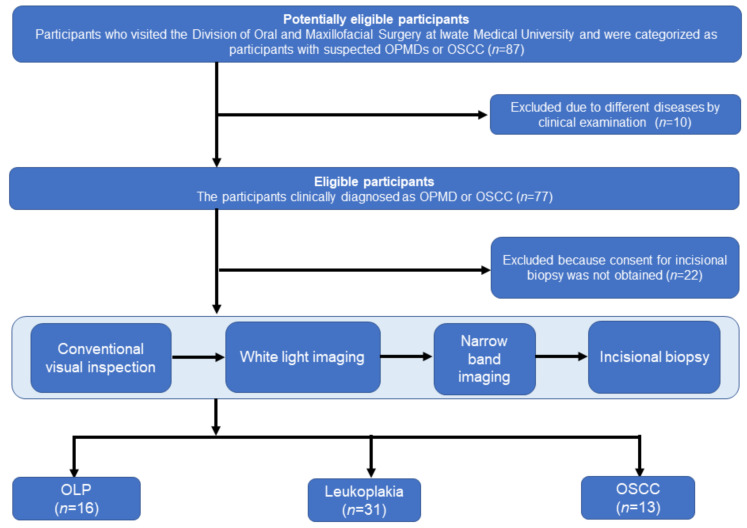
Flow diagram of the participants. Includes epithelial hyperplasia, oral epithelial dysplasia, and proliferative verrucous leukoplakia.

**Figure 2 cancers-14-02415-f002:**
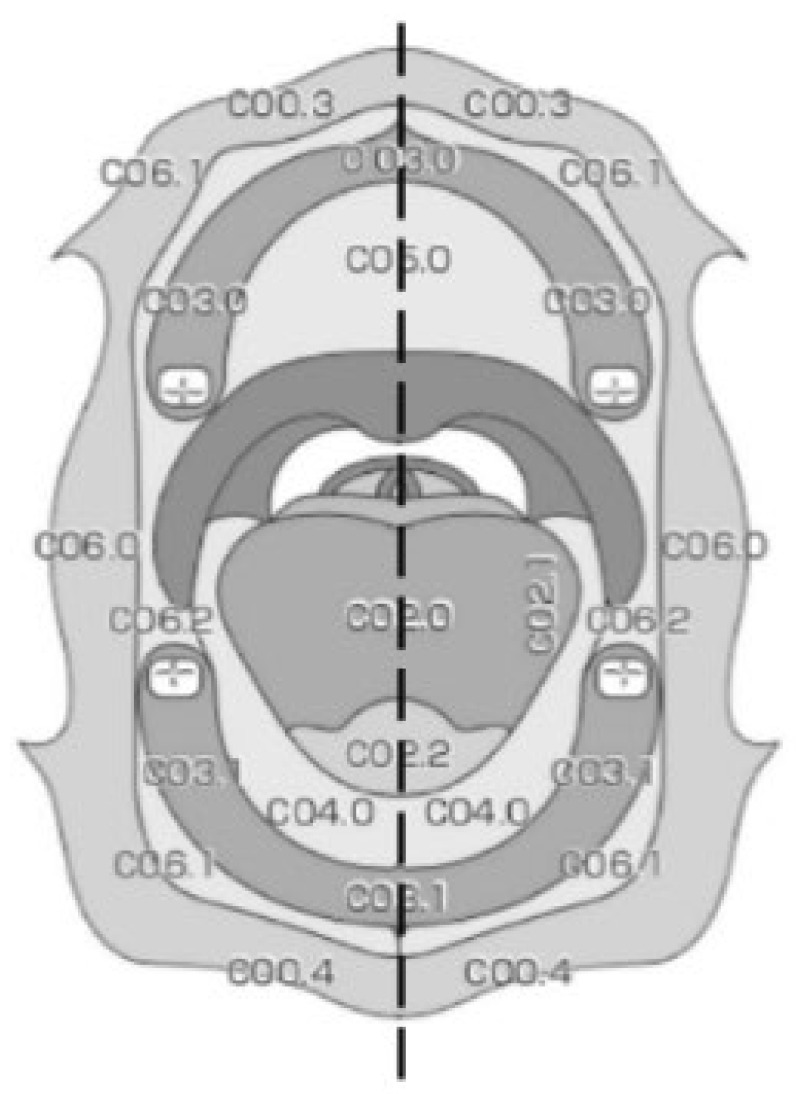
Anatomical sites and subsites of the oral cavity.

**Figure 3 cancers-14-02415-f003:**
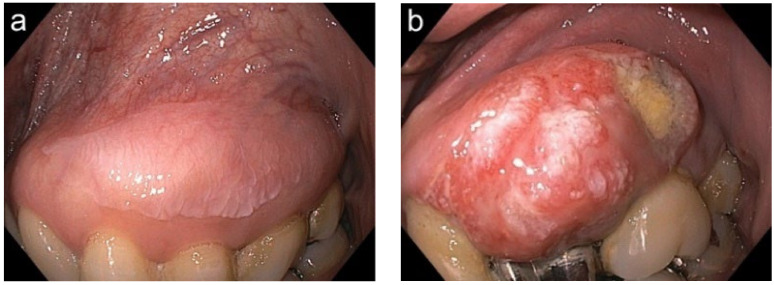
Presentation of a typical homogeneous lesion (**a**), and non-homogeneous lesion (**b**).

**Figure 4 cancers-14-02415-f004:**
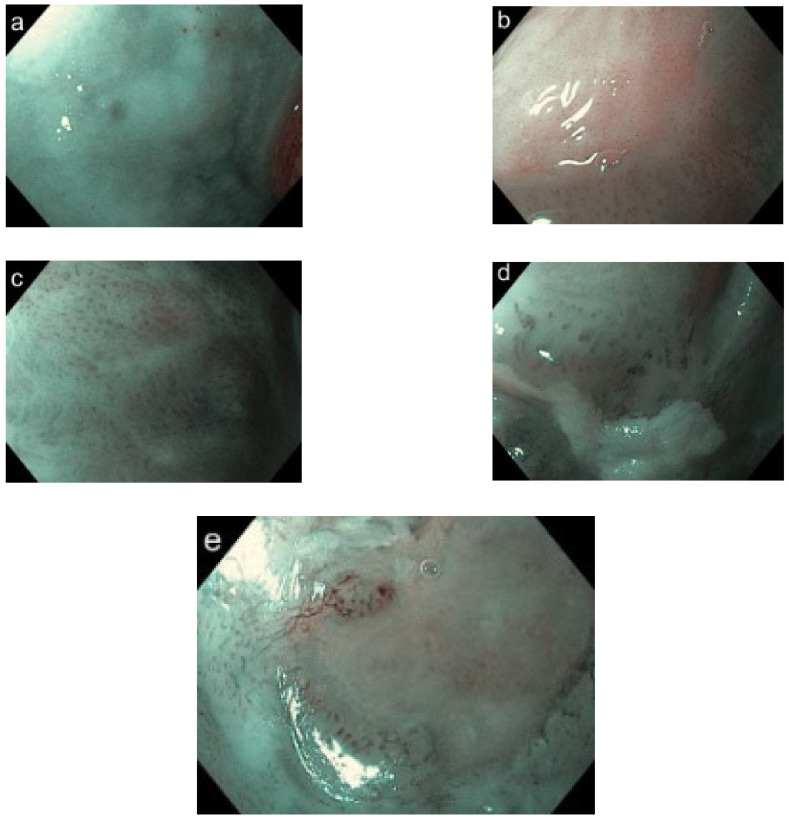
Five types of IPCL classification. Types 0 (**a**), I (**b**), II (**c**), III (**d**), and IV (**e**).

**Figure 5 cancers-14-02415-f005:**
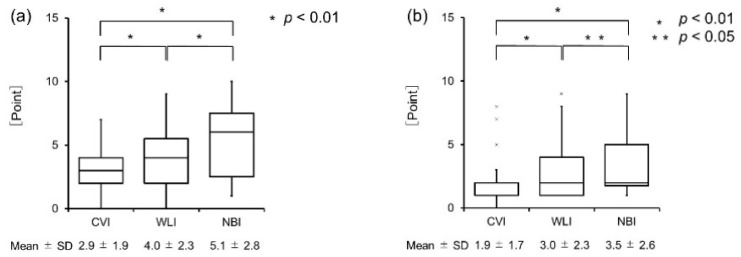
(**a**) Distribution of the OLP lesions by inspection methods. There were significant differences (Wilcoxon’s rank sum test, *p* < 0.01). (**b**) Distribution of the clinical oral leukoplakia lesions by inspection methods. There were significant differences (Wilcoxon’s rank sum test, *p* < 0.01, *p* < 0.05). (**c**) Distribution of the OSCC lesions by inspection methods. There were no significant differences (Wilcoxon’s rank sum test, *p* > 0.01).

**Figure 6 cancers-14-02415-f006:**
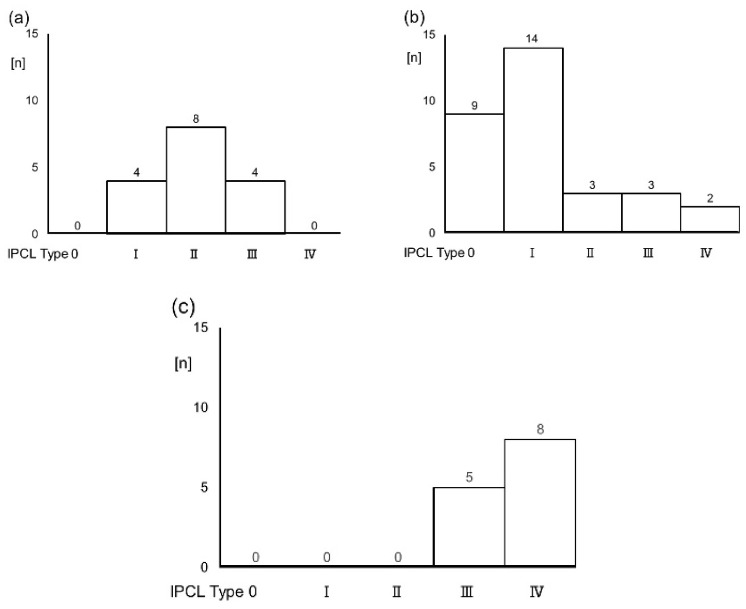
(**a**) OLP lesions were most commonly low-grade (I–II) IPCL patterns (75.0%). (**b**) Clinical oral leukoplakia showed various type of IPCL, including nine (29.0%) type 0, 14 (45.1%) type I, three (9.7%) type II, three (9.7%) type III, and two (6.7%) type IV. (**c**) OSCC were composed of five (38.5%) type III and eight (61.5%) type IV. The distribution of IPCL varied according to the disease. There were statistically significant differences (Kruskal–Wallis test, *p* < 0.001).

**Table 1 cancers-14-02415-t001:** Study cohort demographics, and anatomopathological and clinical characteristics, including the numbers of oral lichen planus (OLP), clinical oral leukoplakia, and oral squamous cell carcinoma (OSCC). The pathological diagnoses of clinical oral leukoplakia were hyperkeratosis (*n* = 1, 3%), epithelial hyperplasia (*n* = 23, 74%), mild epithelial dysplasia (*n* = 2, 6%), moderate epithelial dysplasia (*n* = 3, 10%), and proliferative verrucous leukoplakia (*n* = 2, 6%).

	OLP *n* = 16	Leukoplakia *n* = 31	OSCC *n* = 13
Site	Buccal mucosa *n* = 14 (87.5%) Tongue *n* = 1 (6.2%) Oral vestibule *n* = 1 (6.2%)	Buccal mucosa *n* = 7 (22.6%) Tongue *n* = 9 (29.0%) Hard palate *n* = 5 (16.1%) Gingiva *n* = 10 (32.2%)	Floor of mouth *n* = 1 (7.7%) Tongue *n* = 9 (69.2%) Hard palate *n* = 1 (7.7%) Gingiva *n* = 2 (15.4%)
Clinical subtype			
Homogeneous Non-Homogeneous	*n* = 9 (56.3%) *n* = 7 (43.8%)	*n* = 26 (83.9%) *n* = 5 (16.1%)	*n* = 0 (0%) *n* = 13 (100%)

**Table 2 cancers-14-02415-t002:** Normal intraoral mucosa intraepithelial papillary capillary loop (IPCL) classification. The hard palate and the dorsum of the tongue had thick keratinized mucosa, which made IPCL observation almost impossible. There were statistically significant differences between the groups (Fisher’s exact test, *p* < 0.0001).

Site	IPCL
Hard palate (*n* = 10)	Type 0 80%, Type I 20%
Dorsum of the tongue (*n* = 10)	Type 0 100%, Type I 0%
Gingiva (*n* = 10)	Type 0 40%, Type I 60%
Lips, oral mucosa, oral vestibule, underside of tongue, and floor of mouth (*n* = 10)	Type 0 0%, Type I 100%

**Table 3 cancers-14-02415-t003:** Intra- and inter-observer reliability of the intraepithelial papillary capillary loop classification.

	Intraclass Correlation Coefficient	95% Confidence Interval
Intra-observer	0.81	0.69–0.93
Inter-observer	0.75	0.61–0.88

**Table 4 cancers-14-02415-t004:** Diagnostic accuracy of type III and IV intraepithelial papillary capillary loops by narrow band imaging to determine the possibility of OSCC lesion as a result of incisional biopsy.

Se (95% CI)	Sp (95% CI)	PPV (95% CI)	NPV (95% CI)	Ac (95% CI)
100% (66.1–100)	80.9% (66.7–90.9)	59.1% (36.4–79.3)	100% (86.5–100)	85.0% (73.4–92.9)

Se, sensitivity; Sp, specificity; PPV, positive predictive value; NPV, negative predictive value; Ac, accuracy; CI, confidence interval.

## Data Availability

Not applicable.

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
