# Peer review of "Diagnostic Accuracy of High-Grade Intraepithelial Papillary Capillary Loops by Narrow Band Imaging for Early Detection of Oral Malignancy: A Cross-Sectional Clinicopathological Imaging Study"

_cancers, 2022, doi:10.3390/cancers14102415_

Round 1

Reviewer 1 Report

This paper presents the investigation of the power of narrow-band-imaging (NBI) as a diagnostic tool to detect OSCC (oral squamous cell carcinoma) in a clinical study.

While methods, patients and results are presented well the whole mansucript is quite lengthy. A reduction by at least 1/3 is recommendend and can be achieved easily. So, e.g., the table of the patients' findings offers itself for this.

While the authors speak about a RETROSPECTIVE study it is not clear what the original clinical study was.  Being a retrospective study this implies that all patients have been seen with this technology on a regular way. Why is then a clinical study required? Or is NBI and the diagnostic scheme presented standardyl implemented in clinical routine? If so, this would require explanation of why this investigation is still needed as NBI is implemented (hopefully with appropriate medical justification) in the outpatient ward.

The legend to Fig. 6 will need attention as seemingly the contents of (b) and (a) were confused.

In general, the discussion section of the paper will need extra attention to bring the results in context to the international state-of-the-art of NBI and IPCL literature.

Author Response

Dear referee,

Referee suggested providing a very useful suggestion, so we have now applied the proposed method as possible as we can.

Reviewer:              While methods, patients and results are presented well the whole manuscript is quite lengthy. A reduction by at least 1/3 is recommended and can be achieved easily. So, e.g., the table of the patients' findings offers itself for this.

Author:                 Thank you very much for the suggestion. Indeed, redundant descriptions were found throughout. We have made significant reductions. Also, Table 1 has been rewritten, and a statistical analysis has been added. In addition, raw descriptive patients’ data would be supplemental data.

Reviewer:              While the authors speak about a RETROSPECTIVE study it is not clear what the original clinical study was.  Being a retrospective study this implies that all patients have been seen with this technology on a regular way. Why is then a clinical study required? Or is NBI and the diagnostic scheme presented standardly implemented in clinical routine? If so, this would require explanation of why this investigation is still needed as NBI is implemented (hopefully with appropriate medical justification) in the outpatient ward.

Author:                 The reviewer is certainly right when he discusses the study design. In our institution, we have occasionally used endoscope instruments for the observation of the lesions in the past. However, they have not been used in all cases and there have been clinical questions regarding their usefulness. We have changed the description to cross-sectional study instead of retrospective study. Thank you very much.

Reviewer:              The legend to Fig. 6 will need attention as seemingly the contents of (b) and (a) were confused.

Author:                 I thank the referee for pointing this out. The figure has been revised. Thank you very much.

Reviewer:              In general, the discussion section of the paper will need extra attention to bring the results in context to the international state-of-the-art of NBI and IPCL literature.

Author:                 Thank you very much for pointing this out. We are aware of the truth of your point, but IPCL judgments are quite subjective. It is subject to measurement bias. In this regard, I have changed the title of the article. We have changed the title from Type III or IV IPCL to high grade IPCL. And we have added some description in the text indicating the limitations of this study (line 534-535).

We would like to thank the reviewer once more for sparing the time to write so many detailed and very useful comments. Thank you very much again.

Best regards,

Ikuya Miyamoto

Reviewer 2 Report

  • Despite advancements in cancer treatment, oral cancer has a poor prognosis and is often detected at a late stage. To overcome these challenges, investigators should search for early diagnostic and prognostic biomarkers. More than 700 bacterial species reside in the oral cavity. The oral microbiome population varies by saliva and different habitats of oral cavity. Tobacco, alcohol, and betel nut, which are causative factors of oral cancer, may alter the oral microbiome composition. Both pathogenic and commensal strains of bacteria have significantly contributed to oral cancer. Numerous bacterial species in the oral cavity are involved in chronic inflammation that lead to development of oral carcinogenesis. Bacterial products and its metabolic by-products may induce permanent genetic alterations in epithelial cells of the host that drive proliferation and/or survival of epithelial cells. Porphyromonas gingivalis and Fusobacterium nucleatum induce production of inflammatory cytokines, cell proliferation, and inhibition of apoptosis, cellular invasion, and migration thorough host cell genomic alterations. Recent advancement in metagenomic technologies may be useful in identifying oral cancer-related microbiome, their genomes, virulence properties, and their interaction with host immunity. It is very important to address which bacterial species is responsible for driving oral carcinogenesis. Alteration in the oral commensal microbial communities have potential application as a diagnostic tool to predict oral squamous cell carcinoma. Clinicians should be aware that the protective properties of the resident microflora are beneficial to define treatment strategies. To develop highly precise and effective therapeutic approaches, identification of specific oral microbiomes may be required. discuss and cite doi:10.1177/1533033819867354.
  • the treatment of oral carcinomas is notoriously invasive, often characterized by early relapses. For this reason, in the advanced stages it is often preferred to treat patients with chemoradiotherapy in the first line. However, it has been shown that new robotic approaches allow the treatment of pathologies with radical resections of the tumor. discuss and cite doi:10.1016/j.anl.2021.05.007
  • Despite advances in the surgical management of head and neck squamous cell carcinoma, the identification of synchronous lesions, precancerous lesions around the main tumor, or the unknown primary in the case of neck metastasis remains a problem, as these lesions may be invisible to the naked eye or with standard white light (WL) endoscopy. However, the advent of tools such as narrow-band imaging (NBI) could help the clinician. An interesting study reported at intra-operatively NBI improved the definition of tumor limits in 67.7% of cases, with resection enlargements showing dysplasia and carcinoma in 8 and 12 patients, respectively; the authors obtained 74.2% negative margins at histology. discuss and cite doi:10.1016/j.amjoto.2016.09.020
  • CONSORT diagram could be applied to better describe patients selection.
  • better the table I report mean age, sex rate, location rate, percentage and t test or chi square of the  different location or type.
  • chi square should also be applied in Table II
  • ODD ratio should be calculated for the different findings reported

Author Response

Dear referee,

Referee suggested providing a very useful suggestion, so we have now applied the proposed method as possible as we can.

As you mentioned, there are many different risk factors for oral cancer. We believe it is really important that microorganisms play an important role in carcinogenesis. Our group has previously reported that oral mycoplasmas play an important role in the development of lichen planus and leukoplakia.

doi: 10.3389/fcimb.2017.00403. eCollection 2017.,

doi: 10.1080/20002297.2021.2008153.

The research on oral cancer and microbiology is excellent and we expect to see progress in the future. However, we have omitted our report this time because the insertion of the information about microorganism confuses the reader. I totally agree with the reviewer comments. However, our present study is only an observational study aiming to improve the diagnostic accuracy when using endoscopy for oral cancerous lesion. As the results of this study, an important sign of cancerization is the formation of new abnormal blood vessels. The findings and the microbiological study are very promising. We hope that our study can be considered as a basic study for this purpose.

Regarding the treatment of oral cancer, advanced treatments such as the research that you have presented are very important research. In the present study, when lesions are observed by endoscopy, cancerous lesions can be detected without an invasive biopsy. As well as, our research is not about the treatment or resection of cancer. So, since we did not treat the disease, we cannot make a direct comparison. Therefore, we have not included it in the references. Of course, as you point out, another very important clinical question is whether NBI is useful in determining the extent or safety margin of cancerous lesion. So, your suggestion has been considered and mentioned in the introduction, the determination of the extent of resection with NBI is still a matter of debate. This would be something we would like to discuss in future studies.

The figures used in the CONSORT 2010 statement are mainly reports on randomized control trials (RCTs) and do not match the current study design. We have included figures in line with the STORBE and STARD statements, which are similar checklists. Thank you very much.

Reviewer:             better the table I report mean age, sex rate, location rate, percentage and t test or chi square of the different location or type.

Author:                 Thank you very much for the suggestion. Table 1 has been rewritten, and a statistical analysis has been added in the results section. In addition, raw descriptive patients’ data will be supplemental data.

Reviewer:             chi square should also be applied in Table II ODD ratio should be calculated for the different findings reported

Author:                 Thank you very much for the suggestion. Table 2 has been completely rewritten, and a statistical analysis has been added. After consultation with our university statisticians, the odds ratios were omitted. This was because the small number of cases made statistical analysis impossible.

We would like to thank the reviewer once more for sparing the time to write so many detailed and very useful comments. Thank you very much again.

Best regards,

Ikuya Miyamoto

Round 2

Reviewer 1 Report

Dear authors, 

all my comments have  been answered. The manuscript is fine now.